# Current Perspectives on Biological Screening of Newly Synthetised Sulfanilamide Schiff Bases as Promising Antibacterial and Antibiofilm Agents

**DOI:** 10.3390/ph17040405

**Published:** 2024-03-22

**Authors:** Maria Coanda, Carmen Limban, Constantin Draghici, Anne-Marie Ciobanu, Georgiana Alexandra Grigore, Marcela Popa, Miruna Stan, Cristina Larion, Speranta Avram, Catalina Mares, Mariana-Catalina Ciornei, Aura Dabu, Ariana Hudita, Bianca Galateanu, Lucia Pintilie, Diana Camelia Nuta

**Affiliations:** 1Department of Pharmaceutical Chemistry, Faculty of Pharmacy, Carol Davila University of Medicine and Pharmacy, 6 Traian Vuia Str., 020950 Bucharest, Romania; maria.coanda@drd.umfcd.ro (M.C.); diana.nuta@umfcd.ro (D.C.N.); 2Costin D. Nenitzescu Institute of Organic and Supramolecular Chemistry, 202 B Splaiul Independentei, 060023 Bucharest, Romania; cst_drag@yahoo.com; 3Department of Drug Control, Faculty of Pharmacy, Carol Davila University of Medicine and Pharmacy, 6 Traian Vuia Str., 020950 Bucharest, Romania; anne.ciobanu@umfcd.ro; 4Faculty of Biology, University of Bucharest, Splaiul Independenței 91-95, 050095 Bucharest, Romania; grigore.georgiana-alexandra@s.bio.unibuc.ro (G.A.G.); marcela.popa@bio.unibuc.ro (M.P.); miruna.stan@bio.unibuc.ro (M.S.); 5Research Institute of the University of Bucharest (ICUB), University of Bucharest, Șoseaua Panduri 90, 050663 Bucharest, Romania; larioncristina@yahoo.com (C.L.); ariana.hudita@bio.unibuc.ro (A.H.); bianca.galateanu@unibuc.ro (B.G.); 6National Institute Research and Development for Biological Sciences, Splaiul Independenței 296, 060031 Bucharest, Romania; 7Department of Anatomy, Animal Physiology and Biophysics, Faculty of Biology, University of Bucharest, Splaiul Independentei 91-95, 050095 Bucharest, Romania; speranta.avram@gmail.com (S.A.); catalina.sogor@gmail.com (C.M.); 8Physiology Department, Carol Davila University of Medicine and Pharmacy, 020021 Bucharest, Romania; catalinaciornei@yahoo.com; 9Neurosurgery Department 1, The University Emergency Hospital of Bucharest, Splaiul Independenței 169, 050098 Bucharest, Romania; dabuaura@gmail.com; 10National Institute for Chemical-Pharmaceutical Research and Development, 112 Vitan Av., 031299 Bucharest, Romania; lucia.pintilie@gmail.com

**Keywords:** sulfanilamide, Schiff base, antimicrobial, antibiofilm, cytotoxicity

## Abstract

Growing resistance to antimicrobials, combined with pathogens that form biofilms, presents significant challenges in healthcare. Modifying current antimicrobial agents is an economical approach to developing novel molecules that could exhibit biological activity. Thus, five sulfanilamide Schiff bases were synthesized under microwave irradiation and characterized spectroscopically and in silico. They were evaluated for their antimicrobial and antibiofilm activities against both Gram-positive and Gram-negative bacterial strains. Their cytotoxic potential against two cancer cell lines was also determined. Gram-positive bacteria were susceptible to the action of these compounds. Derivatives **1b** and **1d** inhibited *S. aureus*’s growth (MIC from 0.014 mg/mL) and biofilm (IC from 0.029 mg/mL), while compound **1e** was active against *E. faecalis’*s planktonic and sessile forms. Two compounds significantly reduced cell viability at 5 μg/mL after 24 h of exposure (**1d**—HT-29 colorectal adenocarcinoma cells, **1c**—LN229 glioblastoma cells). A docking study revealed the increased binding affinities of these derivatives compared to sulfanilamide. Hence, these Schiff bases exhibited higher activity compared to their parent drug, with halogen groups playing a crucial role in both their antimicrobial and cytotoxic effects.

## 1. Introduction

Continuous efforts to create novel antimicrobial agents are required due to the escalating resistance observed in microbials towards current medications [1]. Additionally, pathogens that form biofilms present significant challenges in the field of medical care [2]. While traditional antibiotic treatments prove effective against planktonic cells, their impact on biofilm populations, responsible for persistent and device-related infections, is minimal or non-existent [3,4,5].

An economic strategy for developing new therapeutic agents is modifying existing drugs in order to improve their activity or induce new properties. Sulfonamides are a well-known class of synthetic bacteriostatic agents, analogues of *para*-aminobenzoic acid, which competitively inhibit dihydropteroate synthetase (DHPTS), a key enzyme in folic acid metabolism in bacterial cells [6,7]. Their spectrum is large and comprises microorganisms that cannot use folate directly from the environment: both Gram-positive and Gram-negative bacteria, including *Nocardia*, *Actinomyces* spp., *Plasmodium* and *Toxoplasma* species [8]. In recent years, they have received some interest due to their increased resistance to more potent agents [9].

Schiff bases, the condensation products of carbonyl compounds (aldehydes and ketones) with primary amines, have exhibited different biological effects [10], including antibacterial [11] and antibiofilm effects [12]. The >C=N- (imine) bond is polar, capable of forming hydrogen bonds and coordinative interactions. It may serve as a linker between two scaffolds, resulting hybrid molecules, or as a key structural feature of metal complexes, which are important for biological activities [13].

The literature has documented the presence of sulfonamide Schiff bases, either as simple molecules or ligands in metal complexes [14]. Promising results have been obtained from coupling bacteriostatic sulfonamides with different aromatic aldehydes. Returning to benzaldehyde Schiff bases, the designed molecules have been evaluated for their antimicrobial action [15,16,17,18,19] and for their inhibition of biologically relevant enzymes: human carbonic anhydrase (CA) (and its tumor-related isoforms—CA IX, CA XII) [20,21,22], acetylcholinesterase (AChE) [23,24] and urease [25].

The structure–activity relationship of selected compounds investigated for their antimicrobial effect is presented in Figure 1, and those for their anticancer properties in Figure 2.

For antibacterial activity, the effect varied according to the starting amine and the nature and position of the substituents of the aromatic aldehydes (Figure 1). As a general rule, the derivatives exhibited greater potency than their parent drugs and were bactericidal in action. The Schiff bases of 4-aminobenzenesulfonamides are active against Gram-positive cocci, especially *Staphylococci*, including resistant strains and clinical isolates (MIC from 3.91 μM) [15,16,17]. Sulfadiazine Schiff bases may exhibit antifungal and antimycobacterial properties [16]. Mafenide derivatives, on the other hand, have a wider spectrum, comprising Gram-positive and Gram-negative bacteria (MIC from 7.81 μM and 15.62 μM, respectively), mycobacteria (MIC from 3.91 μg/mL), yeast and molds (MIC from 3.91 μM) [19]. Their biofilm inhibition potential has been evaluated in a few cases and it was modest compared with their antibacterial effect [15].

3,5-Dihalogenated-salicylaldehyde moieties are optimal for antibacterial and antifungal activity; the heavier the halogen, the greater the effect. The cytotoxicity of HepG2 varies in the same way [15,16,17,19]. The hydroxyl group in the *ortho* position (X) (Figure 1) is not necessary for antimycobacterial activity, changes in its position and its substitution with acetyl are tolerated [19]. In the case of mafenide derivatives, the exchanging of a benzylidene moiety with 5-nitro-furan or 5-nitro-thiophen is beneficial for their antimicrobial activity, but this increases their cytotoxicity [19]. Substituents of the sulfamoyl group (R) (Figure 1) influenced their activity in a lesser extent; heterocycles like 4,6-dimethylpyrimidine and thiazole were preferred over 5-methylisoxazole [15,18]. The imine bond serves as a linker between the two scaffolds. A reduction of the imine bond or its substitution are detrimental to antibacterial activity. In contrast, substitution is tolerated for a retained antimycobacterial effect [19].

CA IX and CA XII are transmembrane enzymes overexpressed in tumors, such as renal cell carcinoma (CA IX) [26] and breast and brain tumors (CA XII) [27]. Sulfonamides are known inhibitors of these isoforms, as they are able to bind to the Zn^2+^ of the catalytic site of the enzyme [28]. The Schiff bases of different sulfonamides and benzaldehydes have been assayed for their selective inhibition of these isoforms [20,21,22]. The sulfonamide part offers them an affinity to carbonic anhydrase, whereas substituents of benzaldehyde influence their potency and selectivity for a specific isoform (Figure 2) [20,21,22]. A bridge of two atoms is tolerated between the amino group and benzenesulfonamide ring [20]. The imine bond links the two scaffolds. A reduction to amine is tolerated, and even beneficial in some cases, for enzyme inhibition [21]. The imine nitrogen may be implicated in coordinative interactions, with ruthenium complexes being of special interest [29,30]. Regarding the Schiff bases’ mechanism of anticancer action, there is a notable reduction in cell viability, an induction of apoptosis and an up-regulation of reactive oxygen species’ production in cancer cell lines over-expressing CA IX/XII isoforms [31].

**Figure 2 pharmaceuticals-17-00405-f002:**
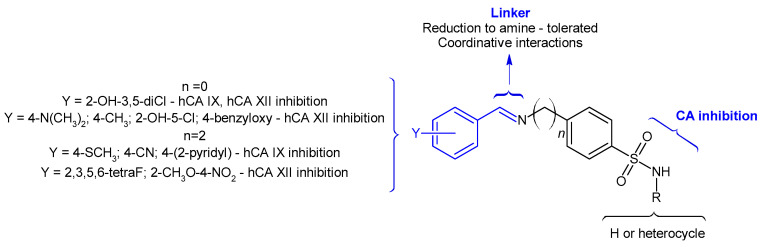
Structure–activity relationship of selected sulfonamide-derived Schiff bases’ anticancer (CA inhibition) effect [20,21,22,29,30].

Taking all of this into account, this study presents the microwave-assisted organic synthesis of five halogenated and non-halogenated sulfanilamide-derived Schiff bases, as well as their spectroscopic and in silico analysis. Their drug- and lead-likeness properties, as well as their ADME (absorption, distribution, metabolism and elimination) profiles, were computed. A comprehensive set of toxicological profiles for the compounds **1a**–**e**, in human and other species, was also determined.

The derivatives were evaluated for their antibacterial and antibiofilm activities against both Gram-positive and Gram-negative strains. Additionally, an in vitro cytotoxicity screening was performed on two cancer cell lines (colorectal adenocarcinoma and glioblastoma). The results were compared with those obtained for the parent drug to understand the impact of derivatization on their pharmacological profile. To elucidate the possible mechanism of action and to identify the relationship between their chemical structure and biological activities (SAR), an in silico analysis was conducted, as well as a docking study against relevant target enzymes.

## 2. Results

### 2.1. Chemistry and Spectral Data

The synthesis of five Schiff base derivatives (**1a**–**e**) of sulfanilamide (**2**) with different substituted benzaldehydes (**3a**–**e**) was performed, with ethanol or methanol as a solvent and glacial acetic acid as a catalyst, under microwave irradiation (Figure 1). TLC and spectral analyses (FT-IR, NMR) confirmed their condensation. The products were characterized by their appearance and melting points.

There are some common features in the IR spectra of the analyzed compounds that confirm their condensation. First of all, they do not exhibit the characteristic bands of an amino group (3500–3300 cm^−1^), which are seen in the spectrum of sulfanilamide. Secondly, bands associated with an azomethine group (1625.7–1619.0 cm^−1^) appeared in the spectra of all derivatives. Corresponding stretching vibrations for NH_2_(SO_2_), an aromatic ring and SO_2_-NH_2_ are present in the spectra of both sulfanilamide and its derivatives (Table 1).

The ^1^H-NMR spectra confirmed the formation of an azomethine bond. All of the compounds presented the characteristic chemical shift of a singlet corresponding to azomethine hydrogen (8.86–8.61 ppm). In ^13^C-NMR’s spectra, the chemical shift of the azomethine carbon appeared in the range of 161.7–157.3 ppm.


**(*E*)-4-{[(4-phenoxyphenyl)methylidene]amino}benzene-1-sulfonamide (1a)**


The compound was prepared using 1 mmol of 4-phenoxybenzaldehyde, 1 mmol of sulfanilamide, 3 mL of methanol and 1 drop of glacial acetic acid. The vial, still closed, was cooled on an ice bath and then in a refrigerator for several hours. The crude product was filtered under vacuum, washed with cold methanol and let to dry. It was recrystallized from methanol. Formula: C_19_H_16_N_2_O_3_S. Molecular weight: 352.41. Appearance: white translucid polyhedric crystals. Yield: 0.267 g (76%). m.p. 162–164 °C. Rf 0.25.

FT-IR (ATR, cm^−1^, solid sample): 3357 w, 3263 w, 1626 w, 1570 m, 1481 m, 1414 m, 1335 m, 1252 s, 1153 vs, 1094 s, 905 m, 881 m, 836 s, 690 vs, 551 vs.

^1^H-NMR (DMSO-d6, δ ppm, J Hz): 8.60 (s, 1H, H-8); 7.97 (d, 8.8, 2H, H-14,10); 7.84 (d, 8.5, 2H, H-4,6); 7.46 (dd, 8.5, 7.6, 2H, H-17, 19); 7.37 (d, 8.5, 2H, H-3,7); 7.33 (s, NH_2_); 7.32 (d, 6.9, 2H, H-10,14); 7.24 (tt, 7.3, 1.7, 1H, H-18); 7.15–7.10 (m, 4H, H-11,13; H-16,20).

^13^C-NMR (DMSO-d6, δ ppm): 161.7 (C-8); 160.2 (C-12); 155.4 (C-15); 154.7 (C-2); 141.0 (C-5); 131.1 (C-10,14); 130.6 (C-9); 130.3 (C-17,19); 126.9 (C-4,6); 124.5 (C-18); 121.2 (C-3,7); 119.7 (C-11,13); 117.9 (C-16,20).


**(*E*)-4-{[(2-bromophenyl)methylidene]amino}benzene-1-sulfonamide (1b)**


The compound was prepared using 2.5 mmol of each reactant (sulfanilamide and 2-bromobenzaldehyde), 3.5 mL of ethanol and two drops of glacial acetic acid. Crystals started to separate after the opening of the vial, and they were filtered and washed with cold ethanol. The solid was recrystallized from ethanol. Obtained spectral data are in accordance with those previously reported [32]. Formula: C_13_H_11_BrN_2_O_2_S. Molecular weight: 339.21. Appearance: white crystals. Yield: 0.610 g (72%). m.p. 164–166 °C (lit. 174–176 °C [32]).

FT-IR (ATR, cm^−1^, solid sample): 3293 w, 3065 w, 1614 m, 1580 m, 1486 w, 1433 w, 1338 s, 1276 m, 1156 vs, 1093 m, 1028 m, 842 s, 828 s, 751 s, 718 s, 557 vs.

^1^H-NMR (DMSO-d6, δ ppm, J Hz): 8.78 (s, 1H, H-8); 8.15 (dd, 7.1, 2.3, 1H, H-11(14)); 7.88 (d, 8.5, 2H, H-4,6); 7.82 (dd, 7.5, 1.7, 1H, H-14(11)); 7.50–7.55 (m, 2H, H-12,13); 7.42 (d, 8.5, 2H, H-3,7); 7.38 (s, 2H, NH_2_).

^13^C-NMR (DMSO-d6, δ ppm): 160.8 (C-8); 154.0 (C-2); 141.7 (C-5); 133.7 (C-9); 133.6 (C-11(14)); 133.4 (C-14(11)); 129.0 (C-12,13); 128.3 (C-13,12); 127.7 (C-4,6); 125.6 (C-10); 121.3 (C-3,7).


**(*E*)-4-{[(2,6-dichlorophenyl)methylidene]amino}benzene-1-sulfonamide (1c)**


The compound was prepared using 2 mmol of each reactant (sulfanilamide and 2,6-dichlorobenzylaldehyde), 3 mL of methanol and 2 drops of glacial acetic acid. The crude mixture was cooled on an ice bath. The vial was opened and the content transferred into a beaker. After the evaporation of the solvent, the yellow semisolid mixture was recrystallized from ethanol. Formula: C_13_H_10_Cl_2_N_2_O_2_S. Molecular weight: 329,20. Appearance: white amorphous solid. Yield: 0.551 g (83%). m.p. 168–171 °C. Rf 0.64; 0.77.

FT-IR (ATR, cm^−1^, solid sample): 3313 m, 3078 w, 1644 s, 1587 m, 1561 m, 1488 w, 1435 m, 1339 vs, 1171 m, 1152 vs, 1093 s, 833 vs, 778 vs, 557 vs.

^1^H-NMR (DMSO-d6, δ ppm, J Hz): 8.75 (s, 1H, H-8); 7.90 (d, 8.8, 2H, H-4,6); 7.52–7.63 (m, 3H, H-11,12,13); 7.39 (s, 2H, NH_2_); 7.38 (d, 8.8, 2H, H-3,7).

^13^C-NMR (DMSO-d6, δ ppm): 158.3 (C-8); 153.6 (C-2); 141.9 (C-5); 134.1 (C-10; C-14); 132.2 (C-12); 131.8 (C-9); 129.2 (C-11, C-13); 127.1 (C-4, C-6); 121.0 (C-3, C-7).


**(*E*)-4-{[(3,5-dichlorophenyl)methylidene]amino}benzene-1-sulfonamide (1d)**


The compound was prepared using 2 mmol of each reactant (sulfanilamide and 3,5-dichlorobenzylaldehyde), 3 mL of ethanol and 2 drops of glacial acetic acid. The reaction mixture was cooled in a refrigerator for several hours. The vial was opened and its contents were transferred into a beaker. After the evaporation of the solvent, the remaining solid was recrystallized from ethanol. Formula: C_13_H_10_Cl_2_N_2_O_2_S. Molecular weight: 329,20. Appearance: white crystalline solid. Yield: 0.609 g (92%). m.p. 171.8–172.7 °C. Rf 0,61.

FT-IR (ATR, cm^−1^, solid sample): 3312 w, 1621 w, 1588 w, 1564 m, 1511 w, 1422 w, 1336 m, 1153 s, 1094 w, 836 w, 664 m, 639 m, 568 m.

^1^H-NMR (DMSO-d6, δ ppm, J Hz): 8.66 (s, 1H, H-8); 7.97 (d, 2.1, 2H, H-10,14); 7.72 (d, 8.5, 2H, H-4,6); 7.83 (t, 2.1, 1H, H-12); 7.42 (d, 8.5, 2H, H-3,7); 7.38 (2H, NH_2_).

^13^C-NMR (DMSO-d6, δ ppm): 160.3 (C-8); 153.4 (C-2); 141.9 (C-5); 138.9 (C-9); 134.7 (C-11,13); 131.0 (C-12); 127.2 (C-10,14); 127.0 (C-4,6); 121.4 (C-3,7).


**(*E*)-4-{[(2,3,5-trichlorophenyl)methylidene]amino}benzene-1-sulfonamide (1e)**


The compound was prepared using 1 mmol of each reactant (sulfanilamide and 2,3,5-trichlorobenzaldehyde), 3 mL of methanol and 1 drop of glacial acetic acid. After the reaction, a white solid was formed. The mixture was first cooled on an ice bath and then put in a refrigerator. The reaction vial was opened and the product was filtered under vacuum, washed with cold methanol and then let to dry. Formula: C_13_H_9_Cl_3_N_2_O_2_S. Molecular weight: 363,65. Appearance: white amorphous solid. Yield: 0.290 g (80%). m.p. 207–209 °C. Rf 0.57; 0.63.

FT-IR (ATR, cm^−1^, solid sample): 3325 w, 3245 w, 3068 w, 1619 w, 1551 w, 1412 w, 1330 m, 1155 s, 1097 m, 889 w, 828 m, 754 m, 527 m.

^1^H-NMR (DMSO-d6, δ ppm, J Hz): 8.86 (s, 1H, H-8); 8.10 (d, 2.6, 1H, H-14(12)); 8.09 (d, 2.6, 1H, H-12(14)); 7.89 (d, 8.5, 2H, H-4,6); 7.48 (d, 8.5, 2H, H-3,7); 7.39 (2H, NH_2_).

^13^C-NMR (DMSO-d6, δ ppm): 157.3 (C-8); 153.0 (C-2); 141.9 (C-5); 135.6 (C-10); 133.8 (C-9); 132.6 (C-14); 132.1 (C-11,13); 127.4 (C-4,6); 126.6 (C-12); 121.6 (C-3,7).

### 2.2. In Silico Studies

#### 2.2.1. Assessment of the Compounds’ Drug- and Lead-Likeness Features

The results generated from the medicinal chemistry filtering (Lipinski, Ghose, Veber and Egan) analyses and bioavailability scoring are presented in Table 2. For the mentioned compounds, their physico-chemical properties, such as their hydrophobicity, an account of their hydrogen bond donor/acceptor atoms, an account of the number of rotatable bonds they have, and their polar molecular surface were computed.

#### 2.2.2. Pharmacokinetics and Pharmacogenomics Profiles of Molecules **1a**–**e**

Furthermore, the ADME-predicted properties of compounds **1a**–**e** were evaluated (Table 3), with emphasis on (i) their human oral bioavailability, (ii) blood–brain barrier (BBB) and central nervous system (CNS) permeability, and (iii) inhibition of renal organic cation transporter 2 (OCT2) and organic anion transporters (OATP1B1, OATP1B3). Also, in Table 3, the pharmacogenomic profiles of compounds **1a**–**e** are mentioned.

The predicted toxicological profiles of synthesized Schiff bases **1a**–**e,** in human and other species, are presented in Table 4.

#### 2.2.3. Computational Pharmacodynamic Profiles of Molecules **1a**–**e**

From the SuperPred database, the most significant molecular targets for compounds **1a**–**e** were extracted. The molecular targets with a model accuracy greater than 70% and a probability of interaction of at least 98% were selected (Table 5).

#### 2.2.4. Molecular Docking

In order to understand the possible mechanisms of antimicrobial action for Schiff bases **1a**–**e**, a docking study was performed on two dihydropteroate synthetases: *E. coli* (PDB ID 1AJ0), a ternary complex with sulfanilamide (Appendix A) [35], and *S. aureus* (PDB ID 1AD4), complex with the natural ligand hydroxymethylene-pterin-pyrophosphate (Appendix A) [36].

The calculated docking scores of Schiff bases **1a**–**e** are presented comparatively in Figure 3.

For the most potent antibacterial compounds (**1b**, **1d**, **1e**), their interactions with the active site of the amino acid residues of selected enzymes are shown in Figure 4 and Figure 5.

### 2.3. Antibacterial and Antibiofilm Screening

An antibacterial assay of the synthesized Schiff bases was performed both qualitatively (agar diffusion method, Appendix A) and quantitively (minimum inhibitory concentration, MIC) on Gram-positive and Gram-negative bacterial strains (Figure 6, Appendix A). Their antibiofilm potential was determined measuring the MIC values of the bacterial adherence to inert substrata (Figure 7, Appendix A).

Compounds **1c** and **1d** exhibited limited inhibition zone formation around the inoculation site for the tested Gram-positive bacteria, whereas the other derivatives had no impact on bacterial proliferation (Appendix A).

Regarding quantitative evaluation, *S. aureus* was the most susceptible tested strain, with MIC values from 0.009 to 0.078 mg/mL. The Schiff bases also exhibited more significant activity than sulfanilamide against *E. faecalis*, with compound **1e** being the most potent (MIC 0.156 mg/mL). In the case of Gram-negative bacteria, their activity was reduced. All tested substances inhibited *E. coli* at the same concentration, 0.625 mg/mL, whereas, on *P. aeruginosa*, the Schiff bases were less active than their parent amine (Figure 6, Appendix A).

The tested compounds were able to inhibit bacterial adherence at concentrations equal to or two times higher than their MIC. A trend was conserved, with *S. aureus* being the most susceptible (Figure 7, Appendix A).

### 2.4. Cytotoxicity Screening

The anticancer potential of the new compounds was investigated, in terms of their cytotoxicity, on HT-29 adenocarcinoma cells and LN229 glioblastoma cells (Appendix A). To assess this, a spectrophotometric MTT assay was employed and the resulting data were statistically analyzed and graphically represented, as seen in Figure 8 and Figure 9, using GraphPad Prism 6.0. Software.

Appendix A presents the first (lowest) doses that induced a statistically significant decrease in the cells’ viability in both cell lines at 24 h and 48 h.

## 3. Discussion

Compounds **1a**–**e** comply with drug-likeness rules (Table 2), indicating their potential drug effect and good bioavailability. Bioavailability is a crucial indicator in drug absorption. All compounds recorded a bioavailability of 0.55, which means that, at a physiological pH, 55% of the compound is expected to reach circulation in an unchanged or active form [37].

The summary of pharmacokinetic profiles presented in Table 3 revealed that (i) all compounds **1a**–**e** exhibited excellent intestinal absorption, (ii) a very good BBB permeability (log BBB varied from −0.90 to −0.55) and (iii) practical CNS permeability, varying from −2.16 to −1.94. Among the analyzed Schiff bases, compound **1b** presents suitable permeability for the BBB, while derivative **1e** showed good permeability in the nervous system and BBB.

The pharmacogenomic profiles of derivatives **1a**–**e** (Table 3) can be summarized as follows: (i) none of the compounds interact with CYP2D6, (ii) only compound **1a** may be a possible inhibitor of CYP2C9 and (iii) all derivatives may inhibit CYP3A4 and CYP1A2, but they are not substrates. Significant results were recorded for the elimination phase. Having acidic sulfonamide group, compounds **1a**–**e** are inhibitors of the hepatic organic anion transporters (OATP1B1, OATP1B3), but not of the renal (OCT2) or hepatic cation transporters (OCT1).

The toxicity profiles of the compounds (Table 4) indicate that (i) all compounds presented affinities for aromatase and estrogen receptors; (ii) all but **1d** showed an affinity for the glucocorticoid receptor; (iii) all compounds could be associated with mitochondrial and hepatic toxicities, but not reproductive toxicity or eye irritation; and (iv) all halogenated derivatives are indicated to be toxic to the respiratory system. The *p*-phenoxy derivative **1a** is the only one associated with carcinogenicity, mutagenic toxicity, androgen receptor binding and human ether-a-go-go-related gene inhibition.

Regarding their predicted pharmacodynamic profiles (Table 5), the most common molecular targets for Schiff bases **1a**–**e** are the endoplasmic reticulum-associated amyloid beta-peptide-binding protein (ERAB) (**1a**–**d**), carbonic anhydrase XII (CA XII), carbonic anhydrase IX (CA IX) (**1c**–**1e**) and cyclooxygenase-2 (COX-2) (**1a**–**c**, **1e**). These results coincide with the usual targets of sulfonamide-based molecules. Other possibly indicated targets were the dual specificity protein kinase CLK4 for compound **1e**, cyclin-dependent kinase 1 for derivative **1c** and transcription intermediary factor 1-alpha for **1a** and **1b**.

Regarding microbiological screening, the isomers **1c** and **1d** were the only ones that showed a small inhibition zone around the inoculum for Gram-positive bacteria on the disc diffusion test (Appendix A). No effect was observed on the Gram-negative representatives.

According to the antibacterial assay (Figure 6, Appendix A), the Schiff bases exhibited selectivity towards *S. aureus.* Halogen substitution appears to be important for their antimicrobial activity. Derivative **1b** ((*E*)-4-{[(2-bromophenyl)methylidene]amino}benzene-1-sulfonamide) was the most potent of the series (MIC 0.014 mg/mL), followed by **1d** (4-{[(3,5-dichlorophenyl)methylidene]amino}benzene-1-sulfonamide) (MIC 0.019 mg/mL). On the other hand, *E. faecalis* was more susceptible to the action of Schiff bases than to that of sulfanilamide. Compound **1e** ((*E*)-4-{[(2,3,5-trichlorophenyl)methylidene]amino}benzene-1-sulfonamide) was the most active in this case (MIC 0.156 mg/mL). The phenoxy derivative **1a** showed only low activity on *S. aureus* and a moderate effect on *E. faecalis*.

None of the compounds, or sulfanilamide, had any effect on *E. coli*. For *P. aeruginosa*, the derivatives were all less active than sulfanilamide (MIC 0.156 mg/mL), with **1b** and **1d** performing slightly better than the rest (MIC 0.234 mg/mL and 0.312 mg/mL, respectively).

Similar results were observed for the antibiofilm assay (Figure 7, Appendix A). Schiff bases **1b** (MIC 0.029 mg/mL) and **1d** (MIC 0.039 mg/mL) were more potent against *S. aureus*’s bacterial adherence than sulfanilamide (MIC 0.078 mg/mL). Compound **1e** was active against the *E. faecalis* biofilm (MIC 0.234 mg/mL), surpassing its parent amine (MIC 0.938 mg/mL). In contrast, on the *P. aeruginosa* biofilm, sulfanilamide performed better than its derivatives (MIC 0.234 mg/mL), with **1b** and **1d** being the most potent (MIC 0.312 mg/mL).

The preference of the compounds for Gram-positive bacteria over Gram-negative ones is not surprising. *P. aeruginosa* is naturally resistant to sulfonamides as it contains efflux pumps [3]. Other Gram-negative bacteria usually acquire resistance through mutations in folP, the gene responsible for the codification of DHPTS, which leads to altered enzymes that have low or no affinity to sulfonamides, thus bypassing the drug [4,38]. Four types of sulfonamide resistance genes, all plasmid-borne, have been discovered in Gram-negative bacteria: sul1, which is present in the class 1 integron; sul2, which is integron independent; sul3, which may be integron-linked [38,39]; and sul4, which is present in *E. coli* [40].

The cytotoxicity screening revealed that the Schiff bases generated a significant decrease in cell viability at low doses both on HT-29 colorectal adenocarcinoma cells as well as on LN229 glioblastoma cells, indicating their high cytotoxic potential on these cancer cell lines.

The highest toxicity on HT-29 colorectal adenocarcinoma cells was displayed by compound **1d,** which induced a statistically significant decrease in the cells’ viability (****, *p* < 0.0001) at 5 μg/mL after 24 h of exposure and at 1.6 μg/mL after 48 h of exposure (Appendix A, Figure 8). Regarding the LN229 glioblastoma cells, the highest toxicity was displayed by compound **1c,** which induced a statistically significant decrease in the cells’ viability (****, *p* < 0.0001) at 5 μg/mL after 24 h of exposure and at 1.6 μg/mL after 48 h of exposure (Appendix A, Figure 9). The same compounds are also indicated to be CA IX and CA XII inhibitors through molecular target prediction. Sulfanilamide was the least cytotoxic in both determinations.

The docking study on *E. coli* DHPTS revealed the increased binding affinities of Schiff bases **1a**–**e** compared to co-crystallized sulfanilamide (Figure 3). The latter interacts with amino acids of the active site primarily through hydrogen bonds: its sulfamoyl group with Arg63 and Ser219 and its amino group with Thr62 (Appendix A). Schiff base derivatives form hydrogen bonds using their free sulfamoyl group and steric interactions using their benzene rings, e.g., Pro145 (Figure 4 and Figure 5). Their additional benzene ring can contribute to supplementary interactions with the active site, e.g., the phenoxy group of compound **1a** forms hydrogen bonds with Arg63 (Appendix A).

In the case of *S. aureus* DHPTS, the docking scores of our compounds are comparable or lower to those obtained with the natural ligand, pterin-pyrophosphate (Figure 3). Besides their sulfamoyl group, their imine group is also implicated in hydrogen bonds with amino acid Arg52 (Figure 6 and Figure 7). In addition, substituents on the second bezene ring contribute to binding to the active site: the phenoxy group forms hydrogen bonds with Arg239 and steric interactions with Asp84 and Met128 (Appendix A); the chloro group in the *metha* position forms steric interactions with Lys207 (Figure 7), while the *ortho* position is not favorable (**1c**—the lowest scores). These observations are in correlation with the results obtained by in vivo antimicrobial tests. Even though compound **1a** generally had the highest scores among the tested derivatives, it did not exhibit an antibacterial effect.

Combining all of the above, some structure–activity relationships may be drawn. Changing the amino group of the sulfonamide derivatives to azomethine contributes to their antibacterial activity; in some cases, it may even enhance the effect. The imine group is able to form hydrogen bonds with the active site. Halogen substitution proves beneficial for their activity, with the 2-bromo, 3,5-dichloro and 2,3,5-trichloro derivatives demonstrating antibacterial and antibiofilm potential. Additionally, a dichloro substitution on benzene ring appears to enhance their cytotoxicity towards cancer cell lines. Thus, compound **1d** presents the most promising antibacterial and anticancer activity. These findings align with results reported in the literature for similar compounds [15,19,22].

## 4. Materials and Methods

### 4.1. Chemistry

#### 4.1.1. General Information

All reactions were performed under microwave irradiation using a Biotage^®^ Initiator Classic 2.0 (Biotage, Uppsala, Sweden), in sealed 2–5 mL reaction vials, under magnetic stirring and a very high absorbance level. The reagents and the solvents were purchased from Sigma-Aldrich, USA (4-phenoxybenzaldehyde); Sigma-Aldrich, Buchs, Switzerland (2,6-dichlorobenzaldehyde); Sigma-Aldrich, Schnelldorf, Germany (sulfanilamide); Merck Schuchardt, Hohenbrunn, Germany (2-bromobenzaldehyde, 3,5-dichlorobenzaldehyde, 2,3,5-trichlorobenzaldehyde); Chemical Company S.A., Iași, Romania (ethanol, methanol); and Chimopar Tranding SRL, Bucharest, Romania (glacial acetic acid).

For thin-layer chromatography (TLC), the methodology indicated in reference [41] was followed. Glass TLC plates coated with silica gel 60, 20 cm × 20 cm (Merck, Darmstadt, Germany), were used as the stationary phase, while the mobile phase was a mixture of chloroform–methanol (9:1, *v*/*v*) (migration distance of 13 cm). The spots were visualized using an acidified potassium permanganate solution. The reference substances were sulfanilamide, 2,6-dichlorobenzaldehyde, 2,3,5-trichlorobenzaldehyde and 2-bromobenzaldehyde.

The infrared spectra were recorded on a JASCO FT/IR-4200 equipped with ATR PRO 450S (diamond crystal). Absorption maxima were reported in wavenumbers (cm^−1^), using the range 400–4000 cm^−1^, with transmittance recorded on the abscissa. The following abbreviations were used: w (weak), m (medium), s (strong), and vs (very strong).

^1^H NMR and ^13^C NMR spectra were recorded on a Varian Gemini 300BB instrument (Varian Medical Systems, Palo Alto, CA, USA) operating at two frequencies, 300 MHz for proton and 75 MHz for carbon-13 NMR. The solvent used was DMSO-d6. The chemical shifts are reported in parts per million (ppm, δ scale) (internal standard—tetramethylsilane, (CH_3_)_4_Si, TMS) and all coupling constant (J) values are in Hertz (Hz). The following abbreviations were used to explain the multiplicities of the ^1^H signals: s (singlet), d (doublet), t (triplet), m (multiplet), dd (doublet doublet) and tt (triplet triplet). The data order is the following: for ^1^H NMR—chemical shifts, multiplicity, coupling constants, number of protons and signal attribution; for ^13^C NMR—chemical shifts, signal attribution.

Melting points (uncorrected) (m.p.) were determined via an open capillary method using an Electrothermal 9100 apparatus (Bibby Scientific Ltd., Stone, UK).

#### 4.1.2. Synthesis and Characterization of the Compounds

The employed synthesis protocol is an adaptation of one previously described [42]. A clean, dry reaction vial (2–5 mL) with a stir bar was charged with sulfanilamide (1–2 mmol), stoichiometric quantities of benzaldehydes, anhydrous methanol or ethanol (3–3.5 mL) and 1–2 drops of glacial acetic acid. Part of the alcohol was used to disperse the reactants and the rest to wash clean the vial. The vial was capped and placed in the reactor cavity. The mixture was stirred for 5 min and then irradiated at 90 °C for 30 min (level of absorbance—very high). Then, the reaction mixture was cooled on an ice bath and/or left in a refrigerator for several hours. The product was filtered under vacuum and washed with ice-cold ethanol or methanol. In some cases, it was first necessary to evaporate part of the solvent and then to filter. The products were purified via recrystallization, from either ethanol (**1b**–**d**) or methanol (**1a**), in the presence of decolorizing carbon.

Thin-layer chromatography was performed to verify the status of the reaction; 0.2% ethanol solutions of the tested mixtures and references were prepared and a migration distance of 13 cm was set. All benzaldehydes migrated with the front, while sulfanilamide had a retention factor of 0.32.

### 4.2. Computational Strategy

#### 4.2.1. Molecular Modeling of Chemical Structures **1a**–**e**

The 3D structure and Simplified Molecular Input Line Entry (SMILES) file of the compounds were obtained for further bioinformatics and cheminformatics analyses using Spartan’20 software (2022) (Wavefunction, Inc., Irvine, CA, USA) [43]. The energies of the structures were minimized using Forcefield MMFF94x with a gradient of 0.05. After minimization, Gasteiger partial charges were applied to all compounds [44].

#### 4.2.2. Assessment of Compounds’ Drug- and Lead-Likeness Features

To evaluate their drug- and lead-likeness feature, the compounds **1a**–**e** were computed, under medicinal chemistry rules, to Lipinski [45], Ghose [46], Veber [47] and Egan [48] filters using the SwissADME web service [49]. These rules impose the following criteria: (i) the Lipinski rule—the molecular weight should not be more than 500 g/mol, the partition coefficient between octanol and water (Log P(o/w) not more than 5, the number of hydrogen bond acceptors should not exceed 10 and the number of hydrogen bond donors should not exceed 5; (ii) the Ghose filter—the molecular weight should be between 160 and 480 g/mol, the Log P(o/w) should be between −0.4 and 5.6, the molar refractivity should be between 40 and 130 m^3^/mol and the number of atoms should be between 20 and 70; (iii) the Veber rule—the number of rotatable bonds not more than 10 and the total polar surface area should not be greater than 140 Å^2^; and (iv) the Egan filter—the Log P(o/w) should not be more than 5.88 and the total polar surface area should not be greater than 131 Å^2^.

#### 4.2.3. Computational Pharmacokinetics, Pharmacogenomics and Toxicological Profiles of Sulfanilamide Schiff Bases **1a**–**e**

The SMILES files of compounds **1a**–**e** were used to predict their pharmacokinetic and pharmacogenomic profiles (absorption, distribution, excretion) using the pkCSM [34] and admetSAR2.0 database [33]. From a large set of pharmacokinetic and pharmacogenomic items, the most relevant were chosen, expressed as numerical and categorical variables: (i) intestinal absorption; (ii) blood–brain barrier (BBB) permeability; (iii) central nervous system permeability (Log CNS); plasma protein binding; and inhibition of the OCT receptor at the renal level. The potential of the compounds **1a**–**e** to serve as inhibitors or substrates of the cytochromes involved in the metabolism of many drugs, such as CYP2D6, CYP3A4, CYP1A2, CYP2C19 and CYP2C9, was investigated. A significant number of items representing toxicity were analyzed: ames (mutagenesis), carcinogenicity, toxicity to different species (crustacea, bees, fish), nephrotoxicity, hepatotoxicity, cardiotoxicity, mitochondrial toxicity and toxicity to nuclear receptors.

#### 4.2.4. Computational Pharmacodynamic Profiles of Molecules **1a**–**e**

The pharmacodynamic features and possible molecular mechanisms by which compounds **1a**–**e** modulated the human enzymes involved in different disorders were investigated using bioinformatics. In this context, the SuperPred database [50] was used to establish the probability of binding of Schiff bases **1a**–**e** to human enzymes based on logistic regression machine learning models. The binding data and resulting model accuracy are based on 646 human targets. In addition, the therapeutic indications of the predicted targets were identified.

#### 4.2.5. Molecular Docking

Two software were used for this assay: the CLC Drug Discovery Workbench 2.4 (2015) [51] and Molegro Virtual Docker (2019) [52]. The structures of the compounds were prepared using Spartan’20 software [53]. The *E* isomer of the Schiff bases was chosen for docking. The protein structures were imported from the Protein Data Bank (http://www.rcsb.org/ (accessed on 3 January 2024).

The docking protocol was carried out according to each software’s requirements. It can be summarized as follows: extraction of the co-crystallized ligand, identification of the binding site and binding pocket, redocking of the ligand and project validation. The hydrogen bonds between co-crystallized and amino acids’ residues were identified and the group of interactions was established. The new ligands were then introduced into the project, and their interactions and docking scores were determined.

### 4.3. Antibacterial and Antibiofilm Bioactivity Screening

#### 4.3.1. General Information

In order to assess the antibacterial and antibiofilm activity of the tested compounds, the following assays were performed: a qualitative evaluation of their antibacterial activity (agar diffusion method) and quantitative evaluations of minimum inhibitory concentration (MIC) and their minimum inhibitory concentrations against bacterial adherence to inert substrata. Four representative bacterial strains were selected: *Escherichia coli* ATCC 25922, *Pseudomonas aeruginosa* ATCC 27853 (Gram-negative), *Enterococcus faecalis* ATCC 29212 and *Staphylococcus aureus* ATCC 25923 (Gram-positive).

#### 4.3.2. Methods

The qualitative evaluation of the compounds’ antibacterial activity was conducted using the agar diffusion method, following CLSI (Clinical and Laboratory Standard Institute) guidelines. A 0.5 McFarland inoculum was dispersed on Mueller Hinton agar plates, and 5 µL of each compound (at a concentration of 10 mg/mL in DMSO) was placed on the agar surface. Following overnight incubation at 37 °C, the antibacterial activity was quantified as the inhibition zone surrounding the area where the compound was applied, and its diameter was measured using a ruler.

The quantitative evaluation of antibacterial activity was conducted as follows: Serial binary dilutions of the test compounds were prepared in 96-well plates, resulting in concentrations ranging from 5 to 0.009 mg/mL. A bacterial suspension of 106 CFU/mL (colonies forming units) was added. Following overnight incubation at 37 °C, their MICs were determined as the lowest concentration that inhibited bacterial growth, by reading their optical densities at 620 nm using a Multiskan FC Thermo Scientific spectrophotometer (Thermo Scientific, Waltham, MA, USA). The assay was performed in duplicate, and the results were presented as the mean ± standard deviation (SD).

To evaluate the antibiofilm activity of the compounds, after determining their MICs, the contents of the 96-well plates were discarded, and the wells were washed three times with phosphate-buffered saline. The biofilms were then fixed with methanol, stained with a 1% crystal violet solution for 20 min and treated with a 33% acetic acid solution for 15 min. Their absorbance at 492 nm was measured using a plate-reading spectrophotometer (Multiskan FC Thermo Scientific). This assay was also performed in duplicate, and the results were presented as the mean ± SD.

### 4.4. In Vitro Cytotoxicity on Cancer Cells

#### 4.4.1. General Information

The cytotoxic potential of the compounds on cancer cells was investigated in vitro by evaluating the cells’ viability after 24 h and 48 h of exposure to several concentrations of the compounds. For this, the following two cancer cell lines were employed: HT-29 (ATCC^®^ HTB-38™)—colorectal adenocarcinoma cells and LN229 (ATCC^®^ CRL-2611™)—glioblastoma cells. These cells were grown in Dulbecco’s modified Eagle medium (DMEM), supplemented with 10% FBS and 1% penicillin/streptomycin mixture (10,000 units/mL penicillin and 10 mg/mL streptomycin), in standard culturing conditions (37 °C, humidified atmosphere of 80RH and 5% CO_2_).

#### 4.4.2. Methods

Cell viability was quantified by an MTT spectrophotometric assay. Initially the cells were seeded at a density of 103 cells/cm^2^ in flat-bottomed 96 well plates and allowed to adhere for 24 h in complete culture medium under standard culture conditions. The following day, the culture medium was replaced with treatments consisting of the following concentrations of compounds prepared in complete culture medium: 1 mg/mL; 500 μg/mL; 200 μg/mL; 50 μg/mL; 40 μg/mL; 8 μg/mL; 5 μg/mL; and 1,6 μg/mL. Fresh complete culture medium was added on control monolayers. After 24 h and 48 h, respectively, the treatments were removed and the monolayers were incubated at 37 °C for 4 h with a 1 mg/mL MTT (3-(4,5-dimethilthiazol-2-il)-2,5-dipheniltetrazolium bromide) solution freshly prepared in serum-free DMEM. The resulting formazan crystals formed by the metabolically active cells were solubilized in 2-propanol, and the absorbance of the resulting solution was determined at 550 nm using a FlexStation III multimodal reader. The spectrophotometric data were statistically analyzed by applying a two-way ANOVA and Bonferroni test and graphically illustrated using GraphPad Prism 6.0. All the experiments were conducted in triplicate and the results are represented as the mean of three independent experiments (n = 3). All the data are expressed as mean ± standard error of the mean. A *p*-value of ≤0.05 was considered statistically significant.

## 5. Conclusions

A series of five Schiff base derivatives of sulfanilamide was successfully synthesized using microwave irradiation. All the derivatives comply with drug-likeness rules, presenting acceptable predicted pharmacokinetic and toxicologic properties. Proteins such as CA IX, CA XII, COX-2 and ERAB were identified as possible molecular targets of compounds **1a**–**e**.

The Schiff bases were evaluated for their antibacterial, antibiofilm and cytotoxicity activity. Gram-positive bacteria were susceptible to the activity of Schiff bases in all assays, whereas Gram-negative strains remained unaffected. The halogenated Schiff bases exhibited better activity than sulfanilamide. Compounds **1b** and **1d** were the most potent against *S. aureus*’s growth and bacterial adherence, whereas derivative **1e** showed the greatest inhibition of *E. faecalis*’s planktonic and sessile forms.

All Schiff bases presented a statistically significant reduction of HT-29 colorectal adenocarcinoma cells and LN229 glioblastoma cells’ viability, with higher levels of cytotoxicity on the LN229 glioblastoma cells. Moreover, compound **1d** exerted the highest cytotoxic effect on HT-29 colorectal adenocarcinoma cells, while compound **1c** exhibited the highest cytotoxicity towards LN229 glioblastoma cells.

This research aligns with our objective of developing new antimicrobial and antibiofilm agents. The identification of these promising molecules encourages further exploration of the valuable biological potential of sulfonamide-derived Schiff bases.

## Data Availability

Data is contained within the article and Appendix A.

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
