# Peer review of "Current Perspectives on Biological Screening of Newly Synthetised Sulfanilamide Schiff Bases as Promising Antibacterial and Antibiofilm Agents"

_pharmaceuticals, 2024, doi:10.3390/ph17040405_

Round 1

Reviewer 1 Report

Comments and Suggestions for Authors

The introduction must be improved incopating diferent types of sulfonamides, e.g. organic vs organometallic, diferent positions of the fragment in molecules and its biological response.

The fig 1 must be corrected and improved in function of the introduction.

please check the Scheme 1. is not clear and the numeration of the molecules is not logic. 

The purification method are not clear. Also the yields are low for MW conditions. The chemicals aspect of the synthesis must be improved. Also, spectroscopic detalis must be improved.

The main manuscript is very long. Some aspect could be moved to the ESI. 

 In my point of view, the suplementary information file is not formal. The authors must be improve the quality of pictures and avoid scan draft pics. 

Comments on the Quality of English Language

The quality could be improved and also technical explanations could be discussing in extension

Author Response

We would like to thank the referee for carefully reading our manuscript and for giving such constructive comments. This is our response to the observations:

  1. The introduction must be improved incopating diferent types of sulfonamides, e.g. organic vs organometallic, diferent positions of the fragment in molecules and its biological response.

We thank the reviewer for the observation. The introduction was improved with structure-activity relationship (SAR) of antimicrobial and antibacterial of selected relevant sulfonamide Schiff bases. Because the present article does not involve synthesis of metal complexes, a brief mention of them was considered sufficient.

  1. The fig 1 must be corrected and improved in function of the introduction.

We thank the reviewer for the observation. Two figures were added in place of figure 1, one which presents SAR of antimicrobial sulfonamide Schiff bases and the other treating anticancer effects.

  1. Please check the Scheme 1. is not clear and the numeration of the molecules is not logic. 

We are very grateful for these suggestion. Scheme 1 was modified accordingly. The numeration corresponds to NMR assigned numbering of atoms. For clarity, a figure with NMR atom numbering was inserted in supplementary materials.

  1. The purification method are not clear. Also the yields are low for MW conditions. The chemicals aspect of the synthesis must be improved. Also, spectroscopic detalis must be improved.

We thank the reviewer for the observation. The text was thus modified to provide more clarity on the methods of synthesis and purification.

The aim was to obtain a general MW protocol, adapting the conventional thermal methods listed in literature. Some optimizations were also required in certain cases (see compound 1b). The yield was greatly influenced by the purification method.

  1. The main manuscript is very long. Some aspect could be moved to the ESI. 

Table 6, table 7, Figure 2 and Group interactions of Figures 4 to 7 were moved to Supplementary Materials.

  1. In my point of view, the suplementary information file is not formal. The authors must be improve the quality of pictures and avoid scan draft pics. 

Thank you for the observation, the Supplementary Materials were modified accordingly.

Reviewer 2 Report

Comments and Suggestions for Authors

1. Should be given in antibacteiral and anticancer mechanism.

2. Please provide the ultrastructureal changes of antibacterial activity.

Comments on the Quality of English Language

1. Should be given in antibacteiral and anticancer mechanism.

2. Please provide the ultrastructureal changes of antibacterial activity.

Author Response

We express our gratitude to the referee for providing valuable comments. Certainly, the observations made prompt us to continue and deepen these studies.

This is our response to the observations:

 1. Should be given in antibacteiral and anticancer mechanism.

We thank the reviewer for this observation. A brief overview of antibacterial effects of sulfonamide Schiff bases was added in the introduction.

With respect to the anticancer mechanism, the purpose of this preliminary study was to detect the antitumor effect of any of the synthesized compounds by investigating their cytotoxicity. Now that we have confirmed promissing results, we will further develop studies to investigate the cytotoxic mechanism of action, including apoptotic pathways. 

2. Please provide the ultrastructureal changes of antibacterial activity.

Thank you for the observation. This is a preliminary study. The ultrastructural changes study will be included in a further research.

Reviewer 3 Report

Comments and Suggestions for Authors

The authors have written an article entitled “Current perspectives in biological screening of newly synthetised sulfanilamide Schiff bases as promising antibacterial and antibiofilm agents”. The authors successfully synthesized a series of five Schiff base derivatives of sulfanilamide was using microwave irradiation. All derivatives are characterized spectroscopically.  These derivatives are evaluated for antiomicrobial and antibioflims activities. Also, their cytotoxic potential against two cancer cell lines was also determined and well tabulated. This reviewer suggest to accept this article after correcting following points:

1.    Page 6, Table 3. OATB1inhibitor or OAPTP1B1 inhibitor and OATB3inhibitor or OAPTP1B3?

2.    Page 18, line 332, (i) all studied Schiff bases may develop mutagenic toxicity and possible androgen receptor binding substrate. But in page 7, Table 4. The derivatives 1a shows different results than others.

3.    Page 18, line 342. 1a, 1b, 1e must be 1a-1c, 1e.

4.    Page 19, Line 385. Table 6 must be table 7.

5.    Page 20, Line 463 …and washed with sold pure solvent. Which solvent was used to wash?

Author Response

We thank the referee for the valuable comments to which we tried to respond and complete the article so that its scientific level increases. This is our response to the observations:

  1. Page 6, Table 3. OATB1inhibitor or OAPTP1B1 inhibitor and OATB3 inhibitor or OAPTP1B3?

Thank you for the observation, it was corrected (OATP1B1, OATP1B3).

  1. Page 18, line 332, (i) all studied Schiff bases may develop mutagenic toxicity and possible androgen receptor binding substrate. But in page 7, Table 4. The derivatives 1a shows different results than others.

Thank you very much for the observation. The paragraph was corrected according to the data in table 4.

  1. Page 18, line 342. 1a, 1b, 1e must be 1a-1c, 1e.

Thank you for the observation, it was corrected

  1. Page 19, Line 385. Table 6 must be table 7.

Thank you for the observation. The table was moved to supplementary materials.

  1. Page 20, Line 463 …and washed with sold pure solvent. Which solvent was used to wash?

We thank the reviewer for the observation. For purification it was used either methanol or ethanol. The phrase was corrected.

Round 2

Reviewer 1 Report

Comments and Suggestions for Authors

Thanks for incorporate the suggestions and improve the quality of the MS

Comments on the Quality of English Language

ok thanks for check